# Characterizations and the Mechanism Underlying Cryoprotective Activity of Peptides from Enzymatic Hydrolysates of *Pseudosciaena crocea*

**DOI:** 10.3390/foods12040875

**Published:** 2023-02-18

**Authors:** Zhe Xu, Zhixuan Zhu, Maolin Tu, Jiale Chang, Shiying Han, Lingyu Han, Hui Chen, Zhijian Tan, Ming Du, Tingting Li

**Affiliations:** 1Key Laboratory of Biotechnology and Bioresources Utilization, College of Life Sciences, Dalian Minzu University, Dalian 116029, China; 2Institute of Bast Fiber Crops & Center of Southern Economic Crops, Chinese Academy of Agricultural Sciences, Changsha 410205, China; 3Key Laboratory of Animal Protein Food Deep Processing Technology of Zhejiang Province, College of Food and Pharmaceutical Sciences, Ningbo University, Ningbo 315832, China; 4Key Laboratory of Marine Fishery Resources Exploitment & Utilization of Zhejiang Province, Hangzhou 310014, China; 5National Engineering Research Center of Seafood, Collaborative Innovation Center of Seafood Deep Processing, School of Food Science and Technology, Dalian Polytechnic University, Dalian 116034, China

**Keywords:** antifreeze peptides, enzymatic hydrolysis, freeze-thaw cycle, protein oxidation

## Abstract

Antifreeze peptides are a class of small molecule protein hydrolysates that protect frozen products from cold damage under freezing or subcooling conditions. In this study, three different Pseudosciaena crocea (*P. crocea*) peptides were from pepsin, trypsin, and neutral protease enzymatic hydrolysis. It aimed to elect the *P. crocea* peptides with better activity through molecular weight, antioxidant activity, and amino acid analysis, as well as to compare the cryoprotective effects with a commercial cryoprotectant. The results showed that the untreated fillets were prone to be oxidized, and the water-holding capacity after freeze-thaw cycle decreased. However, the treatment of the trypsin hydrolysate of *P. crocea* protein significantly promoted the water-holding capacity level and reduced the loss of Ca^2+^-ATP enzyme activity and the structural integrity damage of myofibrillar protein in surimi. Moreover, compared with 4% sucrose-added fillets, trypsin hydrolysate treatment enhanced the umami of frozen fillets and reduced the unnecessary sweetness. Therefore, the trypsin hydrolysate of *P. crocea* protein could be used as a natural cryoprotectant for aquatic products. Hence, this study provides technical support for its use as a food additive to improve the quality of aquatic products after thawing and provides a theoretical basis and experimental foundation for the in-depth research and application of antifreeze peptides.

## 1. Introduction

In recent years, *Pseudosciaena crocea* has become a widely consumptive fish in China, with a gradually increasing cultured scale. However, with the increasingly prominent problems of scale expansion, the oversupply trend of surimi product also appeared [1]. Bioactive peptides derived from *P. crocea*, including antioxidant peptides, antibacterial peptides, and flavour peptides, etc., have been utilized in the field of food and pharmaceuticals [2,3].

Aquatic products, which have a fresh and tender taste, including high-quality protein, eicosapentaenoic acid, and docosahexaenoic acid, are popular among consumers all over the world. Farvin et al. reported that aquatic products showed a neuroprotective effect, such as intelligence improvement, as well as prevention of cardiovascular diseases and stroke [4]. Frozen storage and cold chain transportation were the two most effective methods to enhance the shelf life of aquatic products. However, protein deterioration was inevitable because of the temperature fluctuations and the shortage of cryogenic technique, which led to the growth of ice crystals and cell damage [5]. Thus, the addition of cryoprotectant revealed a positive effect of this problem [6].

At present, the commonly used commercial cryoprotectants were categorized mainly into phenols, phosphates, sugars, and proteins [7]. Frozen storage of aquatic products treated with antifreeze could improve Ca^2+^-ATPase activity, increase the pH value, inhibit the growth of ice crystals, and reduce mechanical damage of muscle tissue [8]. Among them, 4% sucrose had already become a general commercial cryoprotectant for aquatic products [9]. However, the disadvantages, such as excessive sweetness, high calories, and large dosage, limited the application of cryoprotectants. Thus, many studies have focused on the antifreeze proteins, which are extracted mainly from fish, insects, bacteria, plants, and other organisms growing under extremely cold and high-altitude conditions [10,11]. 

Antifreeze peptides were obtained mainly from food-derived protein sources through specific enzymatic hydrolysis sites, with controllable and efficient preparation characteristics [12]. At present, the reported food-derived antifreeze peptides were prepared mostly from processing by-products, such as edible gelatin or animal skin, which have been widely used in the food industry, including in frozen meat and frozen dough [13,14]. Zhang et al. [7] explored the antioxidant, antifreeze, and procoagulant effects of surimi protein hydrolysates with the silver carp surimi. The results showed that the untreated surimi was particularly susceptible to protein degradation and oxidation induced by freezing and rapid deterioration of gel properties. Different pH enzymes had different types of enzymolysis peptides. Acidic, neutral, and alkaline enzymes [15] can obtain more types of peptides, making it easier to screen bioactive peptides. Moreover, compared with 4% sucrose-treated surimi, the 2% trypsin hydrolysate partially substituted for sucrose could effectively delay the oxidation of cysteine. The carbonylation of amino acids, Ca^2+^-ATPase activity loss and destruction of myofibrillar protein structural integrity also increased the initial gel properties of surimi. As a new type of food additive, antifreeze peptides could effectively reduce the formation and recrystallization of ice crystals in cold-chain food, thereby improving the food quality [16]. However, few studies have focused on the application of fish-derived antifreeze peptides in the frozen storage of fish products.

In this study, peptides were prepared by hydrolyzing *P. crocea* proteins with pepsin, trypsin, and neutral protease, and they were characterized. Samples were treated under the induction of freeze-thaw cycle to evaluate the effects of *P. crocea* polypeptides, including the physicochemical properties of turbot muscle, oxidation degree, denaturation degree, and aggregation degree of myofibrillar protein. The study may provide technical support for promoting fish-derived AFPs as food additives to promote the quality of aquatic products after thawing and may provide a theoretical and experimental basis for the in-depth research and application of antifreeze peptides.

## 2. Experimental Materials and Methods

### 2.1. Materials and Reagents 

*P. crocea* and turbot were purchased from a local market for aquatic items. Trehalose was bought from Tianjin Kemiou Chemical Reagent Co., Ltd. (Tianjin, China). Bovine serum albumin (BSA) was obtained from Solarbio Science & Technology Co., Ltd. (Beijing, China). The Ca^2+^-ATPase kit (A070-3), total sulfhydryl content test kit (A063-2), and carbonyl content assay kit (A087) were from Nanjing Jiancheng Bioengineering Institute (Nanjing, China). All other compounds were of the analytical variety. The experiments were prepared for at least three duplicates.

### 2.2. Preparation of Polypeptide from P. crocea

*P. crocea* was decapitated and eviscerated, pulverized into meat chyle, and mixed with purified water at a material-to-liquid ratio of 1:3 (*w*/*v*). Samples were then homogenized with homogenizer (IKAT18, IKA, GER) at 10,000 rpm for 10 min and stirred at 40 rpm for 4 h at 4 °C. The resultant slurry was whirled in a high-speed chilled centrifuge at 1000× *g* for 10 min (H-1850R, Cence, CHN), and the supernatant was centrifuged again under the same conditions. The obtained supernatant was frozen at −20 °C, followed by lyophilization at −60 °C for 12 h in a lyophilizer (Labconco, Kansas City, MO, USA). The relevant enzymolysis conditions refer to the previous method [15] and were slightly modified. The lyophilized powder of *P. crocea* protein was rehydrated with purified water at a feed-to-water ratio of 0.02 mg/mL. Thereafter, pepsin, trypsin, and neutral protease were added at 5000 U/g (relative to the actual protein content of water-extracted crude protein powder). The enzymatic hydrolysis was then performed at pH 2, 40 °C (pepsin), pH 8, 45 °C (trypsin), and pH 7, 50 °C (neutral protease) for 5 h, respectively. The samples were boiled for 10 min to inactivate the enzyme, adjusted to pH 7, and then centrifuged under the above conditions. The supernatant was lyophilized, and three different peptides of *P. crocea* were obtained as lyophilized powders, named P-H (pepsin hydrolysate), T-H (trypsin hydrolysate), N-P (neutral protease hydrolysate), respectively.

### 2.3. Determination of Protein Concentration

Bovine serum albumin (BSA) was used as the standard curve, and the protein concentration was determined by the biuret method [17]. Specific operation steps were as follows: take 0.0 mL, 0.2 mL, 0.4 mL 0.6 mL, 0.8 mL, and 1.0 mL of standard protein solution (10 mg/mL), then add water to 1 mL, add 4 mL of biuret reagent, and mix well. The reaction was carried out in the dark at room temperature for 20 min, and the absorbance value was measured at 540 nm using an enzyme marker (Bio Tek, Shoreline, WA, USA).

### 2.4. Sample Pre-Treatment

Fresh turbots were purchased from a local supermarket (Dalian, China) and sacrificed immediately. The turbots were kept in ice and then transferred to the lab within 1 h. The back muscle of turbots was divided into the following groups: FF (The fresh fillets have just been sacrificed); Control (the fresh fillets were soaked in aqueous solution, which was used as negative control, where the amount is the same as other groups of solvents); PH (the fresh fillets were soaked in 2 mg/mL pepsin hydrolysate); TH (the fresh fillets were soaked in 2 mg/mL trypsin hydrolysate); NP (the fresh fillets were soaked in 2 mg/mL neutral protease hydrolysate); CP (the fresh fillets were immersed in 4% (*w*/*v*) commercial cryoprotectant (4% (*w*/*w*) sucrose), which served as a positive control); TH-CP (the fresh fillets were dipped in 2 mg/mL TH and 4% CP mixed solutions); TH-CP2 (the fresh fillets were soaked in 2 mg/mL TH and 2% CP mixed solutions, which was used as positive control). Each group of fillets was submerged in 4 °C water for 4 h. All experimental groups were frozen at −20 °C for 24 h and then thawed at 4 °C for 12 h. The first freeze-thaw cycle was completed when the overall temperature reached 4 °C. The F-T process was repeated for three times.

### 2.5. Screening of Antifreeze Peptides

#### 2.5.1. Molecular Weight Distribution (MWD) and Amino Acid Content Determination

Each sample was prepared at 1 mg/mL and filtered through a 0.45 μm membrane. The MWD of composite AFPs was determined by using liquid chromatography (Agilent Technologies, Santa Clara, CA, USA).

Each sample was prepared at 5 mg/mL; three volumes of acetone were then added and rested for 1 h. The supernatant was centrifuged at 10,000× *g* for 10 min, and the supernatant was spin-dried with a rotary still. Thereafter, 0.02 M hydrochloric acid (1 mL) was added to reconstitute. The 0.45 μm membrane was used to percolate the solution. Analysis was performed with an amino acid analyzer (L-8900, Techcomp, Shanghai, China).

#### 2.5.2. Measurement of In Vitro Antioxidant Activity

The DPPH, ABTS, hydroxyl radical scavenging rate, and superoxide anion generation rate of each peptide were determined according to the method of Zhuang et al. [18], and the sample solutions were all configured at 1 mg/mL.

#### 2.5.3. Texture Properties Analysis

Each group of samples was cut into 1 × 1 × 1 cm^3^. The hardness, elasticity, cohesion, adhesiveness, and recovery of the samples were analyzed by the texture analyzer (TA-XT Plus, Stable Micro Systems, Godalming, UK). The test parameters were [19]: P/50 probe (diameter 50 mm), rate 1.0 mm/s before test, rate 1.0 mm/s after test, rate 1.0 mm/s after test, compression ratio 30%, and all samples tested 10 sub-parallel.

### 2.6. Determination of Physical Properties

#### 2.6.1. Water-Holding Capacity

Water-holding capacity (WHC) was calculated by using the defrosted loss rate, cooking loss rate, and centrifugation loss rate. With minor modifications, the technique for calculating the loss of thawing was performed according to the method of Xia [20]. The sample was weighed before thawing, recorded as *W*_0_, then thawed and re-weighed as *W*_1_. The formula was as follow:$$\mathrm{Thawing}\;\mathrm{loss}\;\left(\%\right)=\frac{w_{0}-w_{1}}{w_{0}}\;\times \;100\%$$

Approximately 2 g of meat was placed in a 10 mL centrifuge tube lined with filter paper and centrifuged at 1760× *g* at 4 °C for 10 min. *W*_2_ and *W*_3_ were the weights of the turbot samples before and after centrifugation, respectively. The formula was as follows:$$\mathrm{Centrifugal}\;\mathrm{loss}\;\left(\%\right)=\frac{w_{2}-w_{3}}{w_{2}}\;\times \;100\%$$

The thawed samples were cut into pieces and weighed as *W*_4_, then put into a steaming bag and steamed at 85 °C. When the core temperature reached 75 °C, they were removed, dried, and immediately weighed as *W*_5_. The equation was as follow:$$\mathrm{Cooking}\;\mathrm{loss}\;\left(\%\right)=\frac{w_{4}-w_{5}}{w_{4}}\;\times \;100\%.$$

#### 2.6.2. Colour

A colorimeter (CR-400, Konica Minolta, Tokyo, Japan) was used to determine the L* (brightness), a* (redness), and b* (yellowness) values of the muscles before and after freezing and thawing.

#### 2.6.3. Electronic Nose

The thawed turbot sample was added into the automatic sampling bottle to 1/3. The electronic nose (PEN 3, Insent, Atsugi, Japan) was used for detection.

#### 2.6.4. Electronic Tone

The samples were weighted, and 25 mL of pure water was added, and they were homogenized for 1 min. Subsequently, samples were sonicated for 5 min, rested for 30 min, centrifuged (12,000 r/min, 15 min, 4 °C), and the upper grease layer and filter were removed. The lower precipitate was taken, and the above steps were repeated; the two filtrates were combined, and the volume was fixed to 100 mL. Then, 5 mL of constant volume solution was mixed with 75 mL of deionized water in the special injection cup for the electronic tongue. The subsequent analysis was conducted at room temperature with an electronic tongue (SA402B, Insent, Atsugi, Japan).

### 2.7. Extraction of Turbot Myofibrillar Protein

In accordance with the study of Cao et al., the extraction of myofibrillar protein (MFP) method was carried out with minor modifications [21]. The thawed samples were cut into minced meat, added to 4 times the volume of phosphate buffer (PBS) (20 mmol/L, pH 7.0), homogenized at 3000 rpm for 30 s, and repeated 3 times. The mixture was centrifuged at 4 °C for 15 min (4500× *g*), and the supernatant was discarded. The precipitate was taken in PBS, and the above steps were repeated 3 times (dissolution, homogenization, and centrifugation). Then, the precipitate was dissolved in 4 times the volume of PBS (20 mmol/L, 0.6 M NaCl, pH 6.7), homogenized again, centrifuged, and the supernatant was filtered through 4 layers of gauze; the filtrate was crude extracted myofibrillar protein, which was placed at 4 °C for 24 h. The MFP was stored at 4 °C. The filtrate was stored at 4 °C and measured within 24 h.

### 2.8. Determination of Myofibrillar Protein Conformation

#### 2.8.1. Fluorescence Spectroscopy

The endogenous fluorescence spectrum was measured by using the method of Cao and Xiong [22], with slight changes. The MFP from the five groups was diluted to 0.1 mg/mL in PBS solution (20 mM, pH 6.7, including 0.6 M NaCl). A fluorescence spectrophotometer (RF-5301PC, SHIMADZU, Kyoto, Japan) was used to obtain the emission spectrum between 300 and 400 nm, with the excitation wavelength of 295 nm, the slit width of 10 nm, and the sensitivity of 3. Each group of samples was measured for three times.

#### 2.8.2. UV Absorption Spectroscopy

Ultraviolet absorption spectroscopic scanning referred to the method of Xu and Jiang [23], with some slight modifications. The MFPs of the five examples were diluted to 0.5 mg/mL with PBS (20 mM, pH 6.7, 0.6 M NaCl) as blank control group. Results were detected by using the UV-visible spectrophotometer (UV-1600, MAPADA, Shanghai, China), with the scanning detection wavelength range of 200–400 nm.

#### 2.8.3. Circular Dichroism

According to the method of Sun et al. [24], the extracted myofibrillar proteins were diluted with PBS (20 mM, pH 6.7, 0.6 M NaCl) to 0.2 mg/mL for circular dichroism detection. The scanning spectral range was 190–240 nm, the diameter of the cuvette was 1 mm, the measurement temperature was 25 °C, the scanning rate was 50 nm/min, and the response time was 0.25 s. Buffer B was used as a blank sample, and the content of myofibrillar protein secondary structural units was analysed with the software that comes with the instrument.

### 2.9. Determination of MFP Aggregation

#### 2.9.1. Determination of Surface Hydrophobicity of MFP

The surface hydrophobicity of proteins was measured by using the 1-benzine-8-sulfonic acid (ANS) fluorescent probe, as previously described by Han et al. [25]. The MFP of each sample was diluted to 0–1 mg/mL with 20 mM PBS (pH 6.7, 0.6 mol/L NaCl). Then, 4 mL of the protein solution was added to PBS solution (20 mm, pH 7.0, containing 0.6 M NaCl) containing 8 mmol/L ANS. The mixture was kept in the dark for 10 min. Fluorescence intensity of the protein-ANS complex was measured by using a microplate reader, with the measurement parameters: excitation wavelength 390 nm and emission wavelength 470 nm. The fluorescence intensity was plotted as the vertical coordinate and the protein concentration as the horizontal coordinate, and the slope was the surface hydrophobicity.

#### 2.9.2. Particle Size and Zeta Potential

Brookhaven 90 Plus nanoparticle size analyser was used to measure the Zeta potential and particle size of the samples, with some slight modifications based on the method of Beliciu et al. [26]. The MFP of each sample was diluted to 0.02 mg/mL with 20 mM PBS (pH 6.7, 0.6 mol/L NaCl), then filtered with 2 mL of protein solution through a 0.45 μm cellulose acetate membrane (aqueous), which was subsequently ultrasonicated for 1 min to remove air bubbles for the following detection.

### 2.10. Determination of the Degree of Myofibrillar Protein Oxidation

#### 2.10.1. Protein Solubility

The MFP of each group was diluted with PBS (20 mM, pH 6.7, including 0.6 mol/L NaCl) for a concentration of 3 mg/mL, which was then rested at 4 °C for 1 h. After centrifugation for 15 min (4 °C, 4500 × *g*), the supernatant was kept to determine the protein content by the biuret method. The degree of solubility of myofibrillar protein was expressed by solubility, and the solubility was calculated according to the following formula:$$\mathrm{Protein}\;\mathrm{solubility}\;\left(\%\right)=\frac{\mathrm{Protein}\;\mathrm{content}\;\mathrm{in}\;\mathrm{the}\;\mathrm{supernatant}}{\mathrm{Total}\;\mathrm{protein}\;\mathrm{content}\;\mathrm{in}\;\mathrm{the}\;\mathrm{sample}}\;\times \;100\%$$

#### 2.10.2. Total Sulfhydryl (T-SH) Content

The T-SH content was determined according to total sulfhydryl kit (A063-2). The sulfhydryl group reacts with 5,5′-dithio-bis-nitrobenzoic acid (DTNB) to produce a yellow compound with a maximum absorption peak at 412 nm and could be used, therefore, to calculate the total sulfhydryl content.

#### 2.10.3. Ca^2+^-ATPase Activity

The Ca^2+^-ATPase activity was determined according to ATPase kit (A070-3-1). ATP can be hydrolyzed by ATPase to produce ADP and inorganic phosphate (Pi), which can be measured by a simple colorimetric reaction. The magnitude of ATPase activity is an important indicator of the impairment of energy metabolism and function in various cells. The Ca^2+^-ATPase activity is expressed as the amount of inorganic phosphate produced per milligram of tissue protein per minute, and the ATPase activity in a sample was calculated by determining the amount of inorganic phosphate activity (μmol (Pi)/mg (pro)/min).

#### 2.10.4. Carbonyl Content

The carbonyl content was determined according to Carbonyl kit (A087-1-1). The carbonyl group reacts with 2,4-dinitrophenylhydrazine to form red 2,4-dinitrophenylhydrazone, which had a characteristic absorption peak at 370 nm and could be used, therefore, to calculate the carbonyl content.

#### 2.10.5. Dityrosine Content

According to the method of Davies et al. [27], the MFP of each sample was diluted to 1 mg/mL with 20 mM PBS (pH 6.7, 0.6 mol/L NaCl), then centrifuged for 5 min (5000× *g*). The wavelengths of excitation and emission were 325 nm and 420 nm, respectively. The measured fluorescence value was used to calculate the dityrosine content (AU).

### 2.11. Statistical Analysis

All experiments were repeated three times in parallel. The experimental results were expressed as values and standard deviations (SD) and compared using one-way ANOVA. Statistical significance (*p* < 0.05) was evaluated with IBM SPSS statistics by Waller-Duncan’s test [27]. The independent variables are the various cryoprotectant classes.

## 3. Results and Discussion

### 3.1. Screening of Antifreeze Peptides

#### 3.1.1. The Composition of MWD and Amino Acids

The MWDs of three polypeptides of *P. crocea* were determined by liquid chromatography (Figure 1a). Notably, the majority of the MWD in three groups were below 3000 Da. Among them, the proportion of components with molecular weight of T-P below 3000 Da was the highest, accounting for an approximate rate of 94.1% (Figure 1b). Molecular weight is a critical parameter which reveals the level of protein hydrolysis [28]. The results showed that three peptides comprised a large percentage of the molecular weight fraction less than 3000 Da, which indicated that collagen interacted with enzymes to produce controlled hydrolysates with antifreeze action. Additionally, data in Table 1 demonstrated that all peptides were rich in hydrophilic amino acid. In contrast, T-P had the highest proportion of hydrophilic amino acids (51.87%), acidic amino acids (8.1%), including 16.46% of the basic amino acids, 34.63% of the essential amino acids, 27.31% of the polar amino acids without charge, 11.31% of the branched chain amino acids, and 6.18% of the aromatic amino acids. Previous studies showed that antifreeze peptides (AFPs) frequently comprised hydrophilic amino acids, such Asp, His, Glu, Lys, Gln, Arg, Ser, Thr, and hydroxyprolinen [29,30]. Furthermore, Wu et al. [31] reported that after ice affinity extraction, a frozen ice peptide derived from sericin included a high concentration of hydrophilic amino acids, including Ser, Thr, Gly, and Ala. The results suggested that these amino acid residues might be coupled with ice crystals to produce antifreeze effects.

#### 3.1.2. In Vitro Antioxidant Properties

The in vitro antioxidant activities of the three peptides were comprehensively evaluated by four indicators. As shown in Figure 2, trypsin hydrolysate illustrated the best effect on radical scavenging, which was significantly (*p* < 0.05) higher than that of pepsin hydrolyzed peptides and neutral protease hydrolyzed peptides. After hydrolysis, peptides were released from their native proteins, which contained functional groups of amino acids, such as indolic, phenolic, and imidazole groups of amino acids, including Trp, Tyr, and His. Those amino acids could serve as proton donors to terminate free radical chain reaction [32]. Moreover, the MW of hydrolysates also determined the antioxidant properties since lower MW peptides had a greater ability to quench free radicals [33].

#### 3.1.3. TPA

The peptide with better antifreeze activity were preliminarily elected through the analysis of texture properties. As illustrated in Table 2, after three freeze-thaw cycles, the hardness, elasticity, cohesion, adhesion, chewiness, and reversibility of the untreated fish were significantly reduced (*p* < 0.05), likely due to the stiffness of the muscle caused by freezing and deterioration, which was consistent with previous studies. However, TH could maintain the reduced change in the qualitative properties (compared with the fresh group) after three freeze-thaw cycles, which was also equivalent to CP. However, the combined with CP suggested an opposite result compared with TP or commercial CP only. The antagonistic effect indicated that trypsin hydrolysate might replace commercial cryoprotectant as an anti-freeze protection agent.

On the basis of the results of molecular weight, amino acid analysis, antioxidant and texture, trypsin hydrolysate was selected to be used in the further study.

### 3.2. Determination of Physical Properties

Table 3 shows the effects of polypeptides on the whiteness and WHC of turbot caused by F-T cycles. The centrifugal loss of control had an increase of 5.16% compared with FF after three cycles F-T. During the F-T cycles, ice crystals continued to grow and recrystallize, causing significant damage to muscle tissue and myogenic fibrous proteins, which, in turn, may have led to a decrease in muscle WHC [34]. Moreover, ice crystal growth led to cell membrane disruption, causing cytosol loss and lipid oxidation, while protein aggregation and denaturation occurred [35]. After three F-T cycles, the thawing loss in the TH group was less than that of the sample without trypsin hydrolysate treatment. The centrifugal loss of the samples followed the same pattern as the thawing loss. Notably, the trypsin hydrolysate had the same effect compared with the commercial cryoprotectant, which had a thawing loss decline of 5.74% compared with the control sample after three F-T cycles.

Similarly, cooking losses were also characterized by the large amounts of heat losses. WHC dropped due to the release of chemically bound water during cooking, which was caused by fat melting and the denaturing of proteins [36]. To reduce cell membrane damage, TH had the potential to alter crystal structures and result in smaller ice crystals [37]. The lightness and redness of the sample without peptide or sucrose treatment was significantly decreased. Similarly, the yellowness increased compared with the FF three F-T cycles, whereas the addition of peptide or sucrose could better inhibit this deterioration, without significant difference. This phenomenon was most likely caused by cell damage and fluid exudation [38]. Moreover, free radicals caused the crosslink between pigment protein and muscle protein, preventing lipid oxidation [39].

The electronic nose is a specific instrument that can obtain the profile of volatile compounds, which could avoid the deficiencies of the human body’s own sense of smell and enhance the stability and repeatability of odor identification. The odor radar maps of different samples were established by extracting the response values of each sensor, and the results are shown in Figure 3. During the F-T cycle, the difference in the response intensity of the W2S sensor was obvious. The volatile substances in aquatic products included mainly aldehydes and ketones, which were also the main source of fishy smell.

The five basic flavors of fish meat after F-T cycles were analyzed by electronic tone. Compared with FF, there was no significant change in saltiness and sourness in each group. For bitterness, TH had a certain degree of increase, which might because of the nature of the peptides itself; however, due to its good anti-freeze protection effect and very little addition, the bitterness was not higher than that of control. For the umami taste, after three freeze-thaw cycles, each group decreased compared with FF, but trypsin hydrolysate had the best effect on maintaining the umami taste of the samples. In terms of sweetness, due to the large amount of sucrose added, the samples in the relevant group still showed a high sweetness value after three freeze-thaw cycles, and this was also the most obvious drawback of commercial cryoprotectants when applied to aquatic products.

### 3.3. Circular Dichroism

Among the secondary structures of proteins, the most ordered and stable was the α-helix structure, while the β-turn and random coil were disordered structures [40]. Figure 4b shows that the control group α-helix content dropped, which was changed into β-sheet, β-turn, and random coil. This phenomenon demonstrated that repeated freeze-thaw cycles ought to promote the transformation of α-helix in myofibrillar protein to other structures, so that the α-conformation was no longer compact, and the arrangement in space could be more extended; the hydrophobic group was exposed, the binding site was exposed, and it was easier to be bound, so the increase of surface hydrophobicity caused the denaturation of protein molecules, thus affecting the biochemical properties of myofibrillar proteins. 

### 3.4. Fluorescence Spectroscopy

Endogenous fluorescence was one of the key technical methods for determining the tertiary conformation of proteins, which only excite tryptophan residues at the wavelength of 295 nm and may be utilized to represent the degree of oxidation of tryptophan residues as well as changes in their microenvironment [41]. The endogenous fluorescence spectra of myofibrillar proteins of the five groups of samples are shown in Figure 4c. Compared with the experimental group, the maximum fluorescence peak position (λmax) of myofibrillar protein from fish in the control group had a blue-shift phenomenon (shifted to the short-wave direction), indicating that during the freeze-thaw cycle, the protein was denatured, the protein molecules were unfolded, and the polarity of the environment in which the tryptophan was located increases, resulting in increased surface hydrophobicity. The other experimental groups had no significant change. At the same time, compared with the control group, the fluorescence intensity of each experimental group increased sequentially. On the one hand, in the freezing process, ice crystals may grow, resulting in a weak fluorescence intensity, which led to the quenching of amino acids on the side chain of fish protein after thawing by oxidation, and protein oxidation led to molecular aggregation and entrapment of more tryptophan residues. On the other hand, protein unfolding and denaturation exposes tryptophan residues that were originally located on the surface, and the exposure of active groups could lead to enhanced intermolecular interactions, resulting in a decrease in endogenous fluorescence intensity. This result was consistent with previous studies [42].

### 3.5. UV Absorption Spectroscopy

Due to the presence of aromatic amino acids, such as tyrosine and tryptophan, in the ultraviolet absorption spectrum, the absorption peaks were superimposed, thus making it difficult to distinguish the peak positions. The UV absorption second-order guide spectrum could superimpose the spectrum caused by the aromatic amino acid residues to separate the peaks, which effectively improved the resolution of the UV absorption spectrum. Therefore, it could be used to analyze the changes in the microenvironment of characteristic amino acid residues and analyze the tertiary conformation of proteins [23].

The UV absorption second-order derivative spectrum of myofibrillar protein had two positive absorption peaks and two negative absorption peaks. The absorption peak at 288.5 nm was attributed to the combined action of tryptophan and tyrosine residues, and the absorption peak at 296.5 nm was attributed only to the action of tryptophan residues. As shown in Figure 4d, compared with the experimental group, the absorption peak at 296.5 nm of the control group showed a blue-shift phenomenon, indicating that after freezing and thawing, the protein molecules were denatured and expanded, and the environment where tryptophan was located increased in polarity, resulting in enhanced surface hydrophobicity. This result was consistent with the fluorescence spectrum. Furthermore, due to the high degree of protein oxidation in the control sample, the degree of protein aggregation was large, resulting in the movement of tyrosine residues to higher polar regions.

### 3.6. Determination of Surface Hydrophobicity of MFP

Surface hydrophobicity increased in all treatment groups compared with FF. As seen in Figure 5a, following a three F-T cycles, SoANS of MFP pretreated with the blank, TH, CP, and TH-CP was found to be increased by 51.8%, 19.9%, 19.6%, and 30.1%, respectively. Among all treatment groups, the sample prepared with the trypsin hydrolysate and the commercial cryoprotectants solution had the lowest SoANS after three F-T cycles, whereas the control sample had the greatest SoANS (*p* < 0.05). The rise in SoANS reflected a decrease in Ca^2+^-ATPase activity (Figure 5f). This revealed that the hydrophobic domain of MFP isolated from surimi was exposed due to conformational changes [40]. The major causes of surimi protein denaturation were the development of ice crystals and the bound water dissociation during frozen storage. It was also discovered that hydration between hydrophilic amino acid residues in peptides and MFP enhanced the fraction of unfrozen water [43]. As a result, trypsin hydrolysate preserved the surface hydrophobicity of the fillet protein, which made a contribution to its hydrophily and cryoprotective properties during preservation.

### 3.7. Zeta Potential and Particle Size

The protein molecular side chain contains a plurality of polar groups, such as carboxyl groups and hydroxyl groups, which could charge the protein surface when the hydrophilic polar groups were exposed to a protein surface in aqueous solution; the amount of charge carried by amino acids on the surface of white matter and the positivity or negativity of the charge affects the protein surface potential, thereby affecting the Zeta potential value of the protein solution [44]. If the absolute value of the Zeta potential (|ζ|) of the system is large, it means that the amount of the protein had a large amount of charge, and the molecular interpretation is mutually exclusive, thereby inhibiting the aggregation of the protein molecules, reducing the effective particle size value, and enhancing the stability of the system. Therefore, the Zeta potential and the effective particle size could be characterized by the stability of the protein solution.

As seen in Figure 5b, |ζ| value of FF, Control, TH, CP, and CP-TH was 2.1, 0.37, 1.10, 1.07, and 0.70 mV, respectively, and the effective particle diameter values were decreased. Therefore, the immersion pretreatment with trypsin hydrolysate might inhibit the protein aggregation and improve the stability of the protein solution system of the thawed fish, which was comparable to the effect of the commercial cryoprotectant.

### 3.8. Carbonyl Content

The well-documented creation of carbonyl compounds from side chains of amino acids is undoubtedly the most notable effect of MFP oxidation caused by metal ions. Since carbonyl concentration represents the level of protein oxidation, protein carbonylation has been the most widely utilized biomarker for protein oxidation [45]. In accordance with Figure 5c, the carbonyl content of the Control, TH, CP, and TH-CP samples rose to 2.120, 1.447, 1.420, and 1.712 nmol/mg prot, respectively, compared with the FF group (0.611 nmol/mg prot). The results indicated that the content of carbonyl of the control sample was significantly higher than other groups (*p* < 0.05), which was due mostly to the development of ice crystals caused by extrusion, distortion, and cell fracture during the F-T process [46]. Significantly, the disruption activated the oxidase, accelerated protein oxidation, and increased myofibril’s carbonyl group content.

### 3.9. Dityrosine Content

Tyr is an amino acid susceptible to oxidation, during which it was attacked by free radicals and formed covalent cross-links with reactive amino acid residues of other proteins, including complexation between two tyrosine residues, resulting in dimeric tyrosine [47]. As a matter of fact, dityrosine content was identified as one of the protein oxidation-specific indicators [48]. As depicted in Figure 5d, the dityrosine content of FF, Control, TH, CP, and TH-CP was 1102.7, 1533.3, 1253.3, 1284, and 1406.7 AU, respectively. The dithionine content was significantly lower in the TH group compared with the control group (*p* < 0.05). The results surface that immersion of the fillets in trypsin hydrolysate reduces protein oxidation and cross-linking. The greater dityrosine level in the control samples was presumably related to the formation of ice crystallites that disrupted the fillets’ model. As a result, during F-T cycles, there were more oxidation, cross-linking, and protein aggregation [20].

### 3.10. T-SH Content

Under the condition of ROS, the sulfhydryl group oxidation could result in a sequence of complicated processes caused by the creation of oxidized compounds, such as sulfonic acid (RSO_3_H), sulfinic acid (RSO_2_H), sulfenic acid (RSOH), disulfide, and disulfone cross-links (RSSR), implying that T-SH content was a reliable indicator of protein oxidation [49]. As shown in Figure 5e, the substance of T-SH was 149.5, 118.5, 129.0, 131.9, and 121.8 μmol/g prot, corresponding to FF, Control, TH, CP, and TH-CP, respectively. After three F-T cycles, the Control group had the greatest sulfhydryl group content, indicating a serious level of protein oxidation in this gathering, inferable from the creation and recrystallisation of ice gemstones during the F-T cycle. As reported by Nian et al. [50], it is known that AFPs can bind to ice crystals and remain stable in the mixed solution, preventing the production and recrystallization of ice particles caused by freeze-thaw loops, as well as changing the shape of ice crystals to promote cellular integrity and prevent protein oxidation and degeneration. As a result, turbot pre-treated with trypsin hydrolysate maintained outstanding quality after F-T cycles. This was consistent with the dityrosine content result.

### 3.11. Protein Solubility

Figure 5e depicts changes in surimi’s protein solubility. To some extent, protein solubility was shown to be lower in all treatment groups compared with the FF but was comparable to the control group. Moreover, the addition of trypsin hydrolysate had a comparable impact as the commercial cryoprotectant. Protein solubility was shown to be significantly lower in the control group, with a decrease of 28.1% (*p* < 0.05). This suggested that trypsin hydrolysate maintained the protein solubility of turbot surimi. Furthermore, the major cause of the reduction in protein solubility in surimi may be protein freezing denaturation mediated by the separation of bound water during freezing stockpiling [51]. Meanwhile, exposure to hydrophobic aromatic amino acids was discovered to trigger protein aggregation which led to a lower solubility via hydrophobic interactions [52]. As a result, trypsin hydrolysate might limit revealing reactive sulfhydryl groups and hydrophobic aromatic amino acids, reducing the formation of disulfide bonds and the hydrophobicity of the surface. The abovementioned data revealed that the technique might avoid protein aggregation and enhance protein solubility.

### 3.12. Ca^2+^-ATPase Activity

The activation of Ca^2+^-ATPase demonstrated the myosin head’s structural integrity, which may have been caused by the denaturation of the myosin head [53]. The sulfhydryl is an oxidized compound in the head of myosin that induced a reduction activity [54]. From Figure 5f, it can be seen that the Ca^2+^-ATPase activity of the FF sample was 2.01 μmolPi/mgprot/hour, while the values of Control, TH, CP, and TH-CP were decreased to 1.26, 1.59, 1.61, and 1.21 μmolPi/mgprot/hour, respectively. The capacity of Ca^2+^-ATPase was higher in TH and CP, indicating that they had the properties of limiting ice crystal damage to filaments of muscle, reducing the level of protein oxidation. The contrast in neither TH and CP was statistically significant (*p* > 0.05). Combined with the trend in sulfhydryl content, it could be concluded that the decrease in Ca^2+^-ATPase activity is due mainly to the oxidation of protein sulfhydryl groups during the freezing process, leading to the decrease in sulfhydryl concentration.

## 4. Conclusions

The trypsin hydrolysate of *P. crocea* protein showed an in vitro free radical scavenging activity. When dipping fish fillets, the hydrolysate effectively delayed the oxidation and aggregation of myofibrillar proteins, stabilized the secondary structure and tertiary conformation, as well as maintained colour, water-holding capacity, and texture properties. In addition, compared with 4% sucrose, hydrolysate treatment could effectively reduce the sweetness of fillets and enhance the umami. Overall, the trypsin hydrolysate of *P. crocea* protein could replace traditional commercial cryoprotectants as a natural cryoprotectant for aquatic products. Furthermore, studies will, in turn, focus on the specific cryoprotective mechanism of the trypsin hydrolysate of *P. crocea* protein.

## Figures and Tables

**Figure 1 foods-12-00875-f001:**
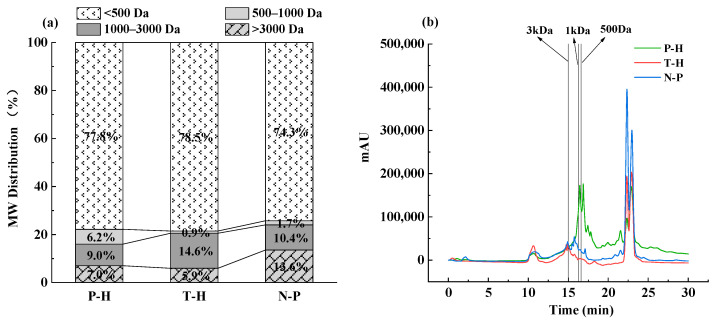
(**a**) Molecular weight distribution and (**b**) liquid chromatogram of three *P. crocea* peptides.

**Figure 2 foods-12-00875-f002:**
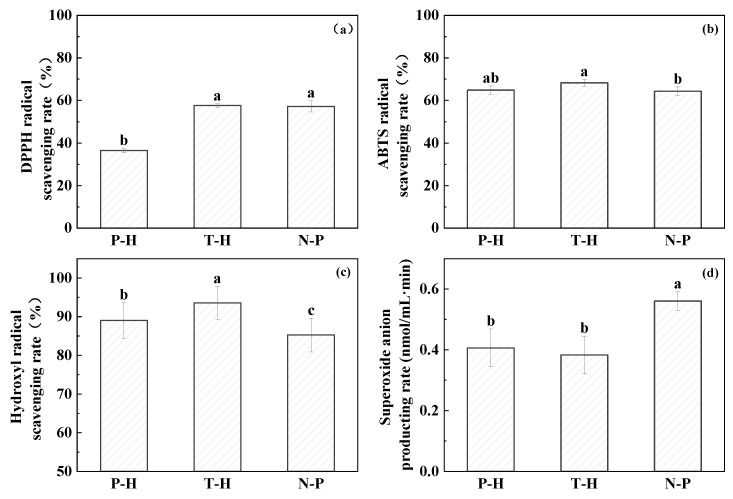
(**a**) DPPH radical scavenging rate, (**b**) ABTS radical scavenging rate, (**c**) hydroxyl radical scavenging rate, and (**d**) superoxide anion production rate of the three peptides. Note: In the significance analysis, the letters from a to c represent the values from large to small, and different letters represent significant differences (*p* < 0.05).

**Figure 3 foods-12-00875-f003:**
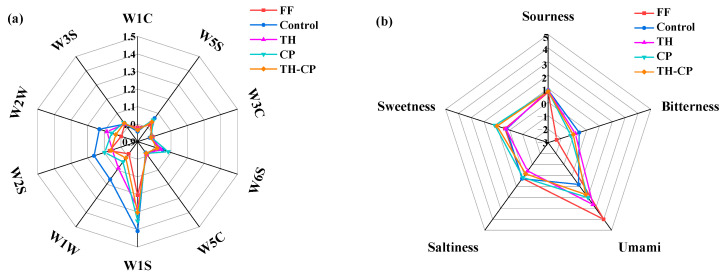
Analysis of (**a**) electronic nose and (**b**) electronic tongue of fillets induced by freeze-thaw cycles with different treatments.

**Figure 4 foods-12-00875-f004:**
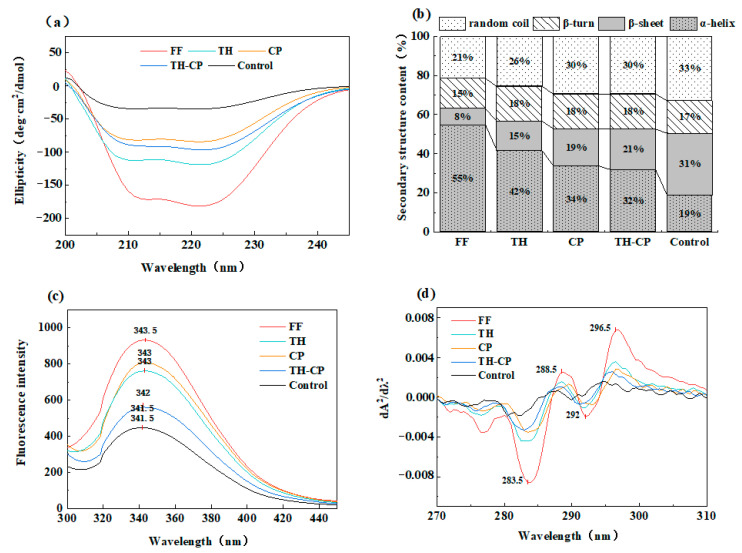
(**a**) Circular dichroism, (**b**) secondary structure content, (**c**) fluorescence spectrum, and (**d**) UV second derivative of myofibrillar proteins induced by freeze-thaw cycles under different treatments.

**Figure 5 foods-12-00875-f005:**
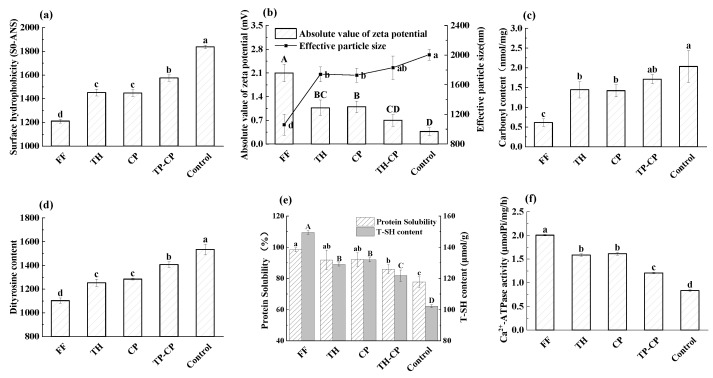
Effects of different treatments on (**a**) the surface hydrophobicity (So-ANS), (**b**) Zeta potential and particle size, (**c**) carbonyl content, (**d**) dimertyrosine content, (**e**) protein solubility, total sulfhydryl content, and (**f**) Ca^2+^-ATPase activity of MFP induced by freeze-thaw cycles. Note: In the significance analysis, the letters from a to d represent the values from large to small, and different letters represent significant differences (*p* < 0.05).

**Table 1 foods-12-00875-t001:** Amino acid composition of three *P. crocea* peptides.

Total Amino Acid	Content (%) P-H	Content (%) T-H	Content (%) N-P
Hydrophilic amino acid ^a^	47.26	51.87	39.14
Polar amino acids without a charge ^b^	24.07	27.31	19.19
Basic amino acids ^c^	18.74	16.46	11.28
Acidic amino acids ^d^	4.46	8.1	8.67
Aromatic amino acids ^e^	3.93	6.18	6.84
Branched chain amino acids ^f^	2.42	11.31	13.22
Essential amino acids ^g^	33.15	34.63	33.23

^a^ Containing Thr, Asp, Ser, Glu, α-AAA, Gly, Cys, Lys, Tyr, His, Ans, and Arg; ^b^ Containing Ser, Thr, Cys, Gly, Tyr, and Asn; ^c^ Containing Lys, His, and Arg; ^d^ Containing α-AAA, Glu, and Asp; ^e^ Containing Phe and Tyr; ^f^ Containing Ile, Leu, and Val; ^g^ Containing Lys, Phe, Trp, Met, Thr, Leu, Ile, and Val.

**Table 2 foods-12-00875-t002:** Effects of different treatments on the texture properties of fish nuggets induced by freeze-thaw cycles.

Samples	Hardness	Elasticity	Cohesion	Adhesion	Chewiness	Reversibility
FF	50.87 ± 9.12 ^a^	1 ± 0.05 ^a^	0.48 ± 0.02 ^a^	24.11 ± 3.19 ^a^	21.68 ± 3.64 ^a^	0.27 ± 0.02 ^a^
Control	35.30 ± 5.68 ^cd^	0.94 ± 0.04 ^c^	0.41 ± 0.04 ^cd^	15.48 ± 2.69 ^c^	15.18 ± 2.49 ^c^	0.20 ± 0.02 ^cd^
PH	35.51 ± 5.51 ^cd^	0.94 ± 0.03 ^bc^	0.41 ± 0.04 ^d^	15.61 ± 2.85 ^c^	15.60 ± 3.12 ^bc^	0.23 ± 0.03 ^b^
TH	44.41 ± 5.5 ^ab^	0.98 ± 0.04 ^ab^	0.46 ± 0.02 ^ab^	20.15 ± 1.18 ^ab^	19.23 ± 1.63 ^ab^	0.23 ± 0.02 ^ab^
NP	32.22 ± 8.08 ^d^	0.94 ± 0.04 ^bc^	0.44 ± 0.03 ^abc^	15.26 ± 2.34 ^c^	15.09 ± 3.01 ^c^	0.22 ± 0.02 ^ab^
CP	45.47 ± 5.62 ^ab^	0.96 ± 0.04 ^abc^	0.45 ± 0.04 ^ab^	18.79 ± 3.74 ^bc^	17.38 ± 4.26 ^bc^	0.19 ± 0.03 ^c^
CP-TH	41.73 ± 5.64 ^bc^	0.93 ± 0.04 ^c^	0.44 ± 0.03 ^bcd^	17.34 ± 4.54 ^bc^	16.11 ± 4.28 ^bc^	0.19 ± 0.02 ^c^
CP-TH2	40.86 ± 5.65 ^bc^	0.93 ± 0.05 ^c^	0.44 ± 0.03 ^bcd^	16.97 ± 4.86 ^bc^	15.73 ± 4.47 ^bc^	0.20 ± 0.02 ^c^

Note: In the significance analysis, the letters from a to d represent the values from large to small, and different letters represent significant differences (*p* < 0.05).

**Table 3 foods-12-00875-t003:** Effects of different treatments on the color and WHC of fillets induced by freeze-thaw cycles.

Samples	L*	a*	b*	Thawing Loss (%)	Cooking Loss (%)	Centrifugal Loss (%)
FF	54.09 ± 0.05 ^a^	2.70 ± 0.17 ^a^	1.14 ± 0.11 ^d^	--	5.19 ± 0.9 ^d^	2.05 ± 0.2 ^d^
TH	50.84 ± 0.89 ^b^	1.31 ± 0.04 ^b^	2.17 ± 0.12 ^c^	6.81 ± 0.42 ^c^	8.23 ± 0.52 ^c^	5.4 ± 0.55 ^c^
CP	49.82 ± 0.96 ^b^	1.28 ± 0.1 ^b^	2.21 ± 0.09 ^c^	6.22 ± 0.79 ^c^	8.74 ± 0.55 ^c^	5.25 ± 0.87 ^c^
TH-CP	46.16 ± 0.95 ^c^	1.14 ± 0.14 ^b^	2.76 ± 0.07 ^b^	9.14 ± 0.61 ^b^	11.11 ± 0.62 ^b^	6.43 ± 0.34 ^ab^
Control	43.09 ± 0.78 ^d^	0.77 ± 0.06 ^c^	3.03 ± 0.18 ^a^	12.55 ± 0.9 ^a^	13.3 ± 0.14 ^a^	7.21 ± 0.59 ^a^

Note: In the significance analysis, the letters from a to d represent the values from large to small, and different letters represent significant differences (*p* < 0.05).

## Data Availability

Data is contained within the article.

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
