# Peer review of "Characterizations and the Mechanism Underlying Cryoprotective Activity of Peptides from Enzymatic Hydrolysates of Pseudosciaena crocea"

_foods, 2023, doi:10.3390/foods12040875_

Round 1

Reviewer 1 Report

This is an interesting manuscript regarding the characterizations and the mechanism underlying cryo-protective activity of peptides from enzymatic hydrolysates of Pseudosciaena crocea. This manuscript is well written and uses many methods and specialized tests to characterize the samples, which provides very useful and valuable information.

I think it is better to mention in the title of the article that the effect of antifreeze was investigated in fillet/surimi. In this case, the article is more related to the food industry.

In the abstract, the following lines need to be re-structured:

“Therefore, P. crocea trypsin hydrolyzed peptides can be used as a natural cryoprotectant for aquatic products, providing technical support for its use as a food additive to improve the quality of aquatic products after thawing, as well as providing theoretical basis and experimental basis for the in-depth research and application of antifreeze peptides.”

In the abstract, why the use of different enzymes for hydrolysis is not mentioned?

In the introduction and the objective part of the manuscript, the following points are ambiguous and need to be explained more clearly: The source of the antifreeze peptides and also the product which these peptides are used in its formulation as well as the proteases using for the protein hydrolysis and the reasons for selecting these enzymes.

Section 2.2, please define the P-P, T-P, and N-P for the first time using in the text.

Section 2.3, protein concentration of what?

In section 2.4, why did you use the peptide solution with a concentration of 2 mg/mL? And please define the F-T process in the text.

For the antioxidant activity tests, did not the turbidity of the samples cause a problem in the testing process? In addition, the DPPH method uses alcohol in its procedure; did this alcohol not precipitate the peptides?

Section 2.8.3, what is the buffer B?

The discussion part of the manuscript should be improved by comparing of the results with more previous studies.

In the figure 2 and table 2, table 3 captions, please define the letters on the means.

The TPA discussion is very superficial and needs to be improved.

Please correct the style of the ref number 9.

The ref 7, 28 and 44, please correct the journal title.

As we know, one of the challenges for using the bioactive peptides in the food formulations is their bitterness, what is the author’s opinion for this issue?

Author Response

Response to Reviewer 1 Comments

This is an interesting manuscript regarding the characterizations and the mechanism underlying cryo-protective activity of peptides from enzymatic hydrolysates of Pseudosciaena crocea. This manuscript is well written and uses many methods and specialized tests to characterize the samples, which provides very useful and valuable information.

I think it is better to mention in the title of the article that the effect of antifreeze was investigated in fillet/surimi. In this case, the article is more related to the food industry.

The authors corrected the manuscript as suggested. Hence, the paper is ready for publication after minor revision, as follows:

Point 1: In the abstract, the following lines need to be re-structured: “Therefore, P. crocea trypsin hydrolyzed peptides can be used as a natural cryoprotectant for aquatic products, providing technical support for its use as a food additive to improve the quality of aquatic products after thawing, as well as providing theoretical basis and experimental basis for the in-depth research and application of antifreeze peptides.”

Response 1: Thanks for the helpful comments. It has been re-structured in the revised manuscript.

Point 2: In the abstract, why the use of different enzymes for hydrolysis is not mentioned?

Response 2: Thanks for the helpful comments. “In this study, three different Pseudosciaena crocea (P. crocea) peptides were from pepsin, trypsin, and neutral protease enzymatic hydrolysis.” has been added to the abstract in the revised manuscript.

Point 3:In the introduction and the objective part of the manuscript, the following points are ambiguous and need to be explained more clearly: The source of the antifreeze peptides and also the product which these peptides are used in its formulation as well as the proteases using for the protein hydrolysis and the reasons for selecting these enzymes.

Response 3: Thanks for the helpful comments. The source of the antifreeze peptides was stated in the INTRODUCTION as “Antifreeze peptides are mainly obtained from food derived protein sources through specific enzymatic hydrolysis sites, with controllable and efficient preparation characteristics. At present, the reported food derived antifreeze peptides were mostly prepared from processing by-products such as edible gelatin or animal skin.” The product which these peptides are used in the formulation and the references was added in the manuscript.

The proteases using for the protein hydrolysis were added in the revised manuscript. The selection of pepsin, trypsin and neutral protease is based on the relatively large range of peptides involved in the enzymatic hydrolysis of protein by acid, alkaline and neutral pH enzymes [1], which makes it easier to screen peptides with antifreeze activity.

[1] Xu, Z., Chen, H., Wang, Z., Fan, F., Shi, P., Isolation and characterization of peptides from mytilus edulis with osteogenic activity in mouse MC3T3-E1 preosteoblast cells. Journal of Agricultural and Food Chemistry 2019, 67(5), 1572-1584.

Point 4: Section 2.2, please define the P-P, T-P, and N-P for the first time using in the text.

Response 4: Thanks for the helpful comments. The relevant details have been added in the revised manuscript.

Point 5: Section 2.3, protein concentration of what?

Response 5: Thanks for the helpful comments. Before the enzymolysis, the protein concentration of P. crocea protein needs to be determined by BSA for the amount of enzyme addition, and in subsequent experiments such as the determination of total sulfhydryl content and the determination of ca-atpase activity, the concentration of myogenic fibrin needs to be determined, and they are all determined by BSA methods.

Point 6:In section 2.4, why did you use the peptide solution with a concentration of 2 mg/mL? And please define the F-T process in the text.

Response 6: Thanks for the helpful comments. It refer to the study of Du et al[1] which investigated the effect of pure antifreeze peptide on the quality of fish meat induced by freeze-thaw cycles and showed that 2 mg/mL of antifreeze peptide was the optimum concentration, so this study was also based on that.

[1] Du, X.; Chang, P.; Tian, J.; Kong, B.; Sun, F.; Xia, X., Effect of ice structuring protein on the quality, thermal stability and oxidation of mirror carp (Cyprinus carpio L.) induced by freeze-thaw cycles. LWT 2020, 124, 109140.

The F-T process was revised as:” All experimental groups were frozen at -20 °C for 24 h and then thawed at 4 °C for 12 h. The first freeze-thaw cycle was completed when the overall temperature reaches 4°C” in the revised manuscript.

Point 7: For the antioxidant activity tests, did not the turbidity of the samples cause a problem in the testing process? In addition, the DPPH method uses alcohol in its procedure; did this alcohol not precipitate the peptides?

Response 7: Thanks for the helpful comments. The sample mixing degree will certainly affect the sample test process. The ethanol used in the DPPH method will produce turbidity in individual samples, but the supernatant is centrifuged for determination during the test process. This leads to the trend of DPPH is not completely the same as other methods, but it can also explain the antioxidant activity of peptides soluble in ethanol of different enzymolysis peptides, and the determination of other methods can more comprehensively characterize the antioxidant activity of enzymolysis peptides.

Point 8: Section 2.8.3, what is the buffer B?

Response 8: Thanks for the helpful comments. Buffer B was PBS (20 mM, pH 6.7, 0.6 M NaCl), and it has been corrected in the revised manuscript.

Point 9: The discussion part of the manuscript should be improved by comparing of the results with more previous studies.

Response 9: Thanks for the helpful comments. The relevant discussion was added in the revised manuscript.

Point 10: In the figure 2 and table 2, table 3 captions, please define the letters on the means.

Response 10: Thanks for the helpful comments. Notes have been added under Figures and Tables in the revised manuscript.

Point 11: The TPA discussion is very superficial and needs to be improved.

Response 11: Thanks for the helpful comments. The analysis has been improved in the revised manuscript.

Point 12: Please correct the style of the ref number 9. The ref 7, 28 and 44, please correct the journal title.

Response 12: Thanks for the helpful comments. They have been corrected in the revised manuscript.

Point 13: As we know, one of the challenges for using the bioactive peptides in the food formulations is their bitterness, what is the author’s opinion for this issue?

Response 13: Thanks for the helpful comments. One of the challenges for using the bioactive peptides in the food formulations is their bitterness, based on this, we performed an electronic tongue analysis and showed that the bitterness of the peptide pretreated samples was not higher than that of the control group after the freeze-thaw cycle, and that the addition was only 2 mg/mL, and was able to overcome the drawbacks of sucrose.

Reviewer 2 Report

In this study, the authors characterized the cryoprotective effects of several enzymatic hydrolysates of Pseudosciaena crocea using turbot as a model. Based on the results obtained, the authors proposed the tryptic hydrolysate of P. crocea as a promising natural cryoprotectant for preserving aquatic products. Strength of the study is that the authors adopted a reasonably broad range of methodologies to analyze both the hydrolysates and the turbot samples in their investigations. I believe findings-wise, the manuscript has significance and needs to be revised (please see feedback listed below). However, please consider rechecking the language/writing to improve grammar, coherence, and clarity.

Below are my feedbacks for the authors’ consideration:

1.     The authors might want to rephrase “trypsin hydrolyzed peptides”, “pepsin hydrolyzed peptides” and “neutral protease hydrolyzed peptides”. These are found throughout the whole manuscript. They seem to suggest that peptides were hydrolyzed, when in reality, the authors hydrolyzed the proteins, not peptides.

2.     INTRODUCTION:

(a)   The 1st paragraph – the last sentence seems unrelated/irrelevant to the first two statements. This is incoherent/confusing. Please recheck.

(b)  The 2nd paragraph – the first statement should be supported with a cited reference.

(c)   Please briefly explain why monitoring Ca2+-ATPase activity is relevant to the objective of this study.

(d)  Please briefly explain why focusing on this species of Pseudosciaena crocea instead of using other fish species as sources of cryoprotective peptides.

3.     M&M:

(a)   In some places, where the past tense should be used, the present tense was used instead. Please check and correct them where appropriate. Examples: “The first freeze-thaw cycle is completed when the overall temperature reaches 4°C.” and “The sample is weighed before thawing…”

(b)  Some sentences are confusing/unclear, for example, “Purified water was reconstituted by adding pepsin…” and “Boiling water was boiled for 10 min to inactivate the enzyme…”.

(c)   Section 2.2

(i)        What was the rationale/basis for using the following hydrolysis parameters: “… pepsin, trypsin as well as neutral protease 5000 U/g (relative to the actual protein content of water-extracted crude protein powder) at a feed-liquid ratio of 1:50 for enzymatic hydrolysis at pH 2, 40 °C (pepsin), pH 8, 45 °C (trypsin), and pH 7, 50 °C (neutral protease) for 5 h”? Were these referred from another study? If so, the reference must be cited. Or were these previously/preliminarily optimized by the authors?

(ii)       For “feed-liquid ratio of 1:50”, what were the units?

(iii)     In the last sentence: “… and three different peptides of P. crocea…” – Rather than calling them “peptides”, I would think it is more accurate to call them “hydrolysates” or “peptide mixtures”.

(d)  Section 2.4 – Please revise this extremely long statement: “The back muscle of turbots was divided into the following groups: The fresh fillets have just
been killed (FF) … which was used as positive control)”
.

(e)   Section 2.5 – Please provide some details for the liquid chromatography parameters used for the determination of MWD.

4.     RESULTS & DISCUSSION:

(a)   Section 3.1.1 – “…indicated that collagen was an essential substrate interacted with alcalase to produce controlled hydrolysates with antifreeze action.” – This part is confusing. Can the authors recheck whether alcalase is relevant to this study? Also, this interpretation would best be supported by a cited reference, or it might sound speculative.

(b)  Table 1 – The data here should be analyzed statistically, like what the authors did for the other tables. Furthermore, please include the standard deviations along with the mean values.

(c)   Section 3.1.2 – “As shown in Fig.2, trypsin hydrolyzed peptides illustrated the best effect on radical scavenging, which was significantly (P<0.05) higher than that of pepsin hydrolyzed peptides and neutral protease hydrolyzed peptides.” – Please recheck/revise this statement. Based on Figure 2, T-P was not always superior to P-P or N-P.

(d)  Section 3.1.3 – Please discuss the significance/relevance of the four parameters “Cohesion”, “Adhesion”, “Chewiness” and “Reversibility” too. Please recheck/revise this statement. Based on Figure 2, T-P was not always superior to P-P or N-P.

(e)   Section 3.2 – “trypsin hydrolyzed peptides had the best effect on maintaining the umami taste of the samples” – Can the authors propose any explanation for this? Is there any supporting information in the literature?

(f)   Section 3.3 – Is there any supporting information in the literature that can be cited to support the interpretation here?

(g)  Sections 3.4 and 3.5 – Are there any studies that can be cited to support the interpretation in the second half of these sections?

(h)  Section 3.8 – “Significantly, the disruption activated the oxidase, accelerated protein oxidation, and increased myofibril’s carbonyl group content.” – This part is unclear. What oxidase was the authors referring to? Also, would it be appropriate then to cite references to support the statement?

(i)    Section 3.9 – “As depicted in Fig. 5(d), the dityrosine content of FF, Control, TP, CP and TP-CP was 1102.7, 1674, 1533.3, 1253.3, 1284 and 1406.7 AU, respectively.” – This part is unclear. Please recheck why there would be six values, not five?

(j)    Section 3.12 – “The contrast in both TP and CP was not statistically significant (P<0.05).” – If it is not statistically significant, it should be P > 0.05, instead. Please recheck/revise.

5.     CONCLUSIONS: Please consider whether it is appropriate/accurate to use the term “peptide” here, e.g., “The P. crocea peptide prepared by trypsin…” and “… the peptide effectively delayed the oxidation”. Since what the authors actually prepared were peptide mixtures or hydrolysates, calling the samples “peptide’ might mislead the readers into thinking pure/purified (single) peptides were used/prepared.

Author Response

In this study, the authors characterized the cryoprotective effects of several enzymatic hydrolysates of Pseudosciaena crocea using turbot as a model. Based on the results obtained, the authors proposed the tryptic hydrolysate of P. crocea as a promising natural cryoprotectant for preserving aquatic products. Strength of the study is that the authors adopted a reasonably broad range of methodologies to analyze both the hydrolysates and the turbot samples in their investigations. I believe findings-wise, the manuscript has significance and needs to be revised (please see feedback listed below). However, please consider rechecking the language/writing to improve grammar, coherence, and clarity.

Below are my feedbacks for the authors’ consideration:

The authors corrected the manuscript as suggested. Hence, the paper is ready for publication after minor revision, as follows:

Point 1: The authors might want to rephrase “trypsin hydrolyzed peptides”, “pepsin hydrolyzed peptides” and “neutral protease hydrolyzed peptides”. These are found throughout the whole manuscript. They seem to suggest that peptides were hydrolyzed, when in reality, the authors hydrolyzed the proteins, not peptides.

Response 1: Thanks for the helpful comments. Three different peptides of P. crocea were named P-H (pepsin hydrolysate), T-H (trypsin hydrolysate), N-P (neutral protease hydrolysate), respectively. They were P. crocea protein hydrolysate, and corrected in 2.2 in the manuscript.

Point 2: (a) The 1st paragraph – the last sentence seems unrelated/irrelevant to the first two statements. This is incoherent/confusing. Please recheck.

Response 2: Thanks for the helpful comments. It has been revised as “In the previous studies of anti-freeze peptides, most of them are insect or plant peptides applied to aquatic products or animal skin peptides applied to dough and other food products, but because the peptides are different from the source of the research object so it has certain disadvantages such as off-flavor, but this study extracted peptides from aquatic products and then applied to aquatic products in turn, which increased the richness of the relevant research. ” in the manuscript.

Point 3: (b) The 2nd paragraph – the first statement should be supported with a cited reference.

Response 3: Thanks for the helpful comments. It has been added the reference 7 [1] in the manuscript.

[1] Zhang, L.; L, Q.; Hong, H.; Luo, Y., Prevention of protein oxidation and enhancement of gel properties of silver carp (Hypophthalmichthys molitrix) surimi by addition of protein hydrolysates derived from surimi processing by-products. Food chemistry Chemistry 2020, 316, 126343.

Point 4: (c) Please briefly explain why monitoring Ca2+-ATPase activity is relevant to the objective of this study.

Response 4: Thanks for the helpful comments. The molecular structure of myosin consists of a head and a tail with Ca2+-ATPase sites in the head, therefore Ca2+-ATPase activity characterizes the structural integrity of the myosin head and its size reflects the denaturation of the myosin head, and the decrease in activity is due to the oxidation of sulfhydryl groups in the myosin head. Therefore, the degree of freezing oxidative denaturation of myogenic fibronectin can be reflected by changes in Ca2+-ATPase. This is explained at 3.12 in the manuscript.

Point 5: (d) Please briefly explain why focusing on this species of Pseudosciaena crocea instead of using other fish species as sources of cryoprotective peptides.

Response 5: Thanks for the helpful comments. P. crocea is an important economic fish in China's offshore, and an important part of people's reasonable dietary structure. In recent years, the farming technology of P. crocea in China has become more and more mature, and the production is increasing rapidly year by year, so there is a certain problem of oversupply, which leads to the waste of resources. Regarding the bioactive peptide made from P. crocea, the antioxidant peptide, antibacterial peptide, taste-presenting peptide, etc. have been used in the field of food and medicine. However, there is not much research on its anti-freezing activity.

Point 6: (a) In some places, where the past tense should be used, the present tense was used instead. Please check and correct them where appropriate. Examples: “The first freeze-thaw cycle is completed when the overall temperature reaches 4°C.” and “The sample is weighed before thawing…”.

Response 6: Thanks for the helpful comments. It has been corrected in the manuscript.

Point 7: (b) Some sentences are confusing/unclear, for example, “Purified water was reconstituted by adding pepsin…” and “Boiling water was boiled for 10 min to inactivate the enzyme…”.

Response 7: Thanks for the helpful comments. It has been corrected in the manuscript.

Point 8: (c) (i) What was the rationale/basis for using the following hydrolysis parameters: “… pepsin, trypsin as well as neutral protease 5000 U/g (relative to the actual protein content of water-extracted crude protein powder) at a feed-liquid ratio of 1:50 for enzymatic hydrolysis at pH 2, 40 °C (pepsin), pH 8, 45 °C (trypsin), and pH 7, 50 °C (neutral protease) for 5 h”? Were these referred from another study? If so, the reference must be cited. Or were these previously/preliminarily optimized by the authors?

Response 8: Thanks for the helpful comments. The relevant enzymolysis conditions refer to the previous method [1] and were slightly modified.

[1] Xu, Z., Chen, H., Wang, Z., Fan, F., Shi, P., Isolation and characterization of peptides from mytilus edulis with osteogenic activity in mouse MC3T3-E1 preosteoblast cells. Journal of Agricultural and Food Chemistry 2019, 67(5), 1572-1584.

Point 9: (c) (ii) For “feed-liquid ratio of 1:50”, what were the units?

Response 9: Thanks for the helpful comments. It has been revised as “The lyophilized powder of P. crocea protein was rehydrated with purified water at a feed-to-water ratio of 0.02 mg/mL.” in the manuscript.

Point 10: (c) (iii) In the last sentence: “… and three different peptides of P. crocea…” – Rather than calling them “peptides”, I would think it is more accurate to call them “hydrolysates” or “peptide mixtures”.

Response 10: Thanks for the helpful comments. It has been revised in the manuscript.

Point 11: (d) I Section 2.4 – Please revise this extremely long statement: “The back muscle of turbots was divided into the following groups: The fresh fillets have just been killed (FF) … which was used as positive control)”.

Response 11: Thanks for the helpful comments. This paragraph is to introduce the 7 experimental groups and their naming abbreviations.

Point 12: Section 2.5 – Please provide some details for the liquid chromatography parameters used for the determination of MWD.

Response 12: Thanks for the helpful comments. It has been added to the section 2.5 in the manuscript.

Point 13: (a) Section 3.1.1 – “…indicated that collagen was an essential substrate interacted with alcalase to produce controlled hydrolysates with antifreeze action.” – This part is confusing. Can the authors recheck whether alcalase is relevant to this study? Also, this interpretation would best be supported by a cited reference, or it might sound speculative.

Response 13: Thanks for the helpful comments. It has been corrected to show that the protein was successfully hydrolyzed to a small molecular weight peptide in the manuscript.

Point 14: (b) Table 1 – The data here should be analyzed statistically, like what the authors did for the other tables. Furthermore, please include the standard deviations along with the mean values.

Response 14: Thanks for the helpful comments. The sum of the standard deviations of each amino acid is meaningless as the standard deviation of the average value. Therefore, the values in the table are the sum of multiple amino acids and are in the form of the average value [1].

[1] Chen X., Wu J., Li X., Investigation of the cryoprotective mechanism and effect on quality characteristics of surimi during freezing storage by antifreeze peptides. Food Chemistry, 2022, 371: 131054.

Point 15: (c) Section 3.1.2 – “As shown in Fig.2, trypsin hydrolyzed peptides illustrated the best effect on radical scavenging, which was significantly (P<0.05) higher than that of pepsin hydrolyzed peptides and neutral protease hydrolyzed peptides.” – Please recheck/revise this statement. Based on Figure 2, T-H was not always superior to P-H or N-P.

Response 15: In DPPH, ABTS and hydroxyl radical scavenging rate, T-H scavenged free radicals better than P-H and N-P. In superoxide anion production rate, T-H was equal to P-H and lower than N-P, indicating that it could inhibit the production of oxygen radicals. Combining the four groups of experiments, T-H had the strongest antioxidant capacity.

Point 16: Section 3.1.3 – Please discuss the significance/relevance of the four parameters “Cohesion”, “Adhesion”, “Chewiness” and “Reversibility” too. Please recheck/revise this statement. Based on Figure 2, T-H was not always superior to P-P or N-P.

Response 16: Thanks for the helpful comments. Significance analysis showed that T-H was the least different from fresh fish in terms of hardness, elasticity, cohesion, adhesion, chewiness and reversibility.

Point 17: (e) Section 3.2 – “trypsin hydrolyzed peptides had the best effect on maintaining the umami taste of the samples” – Can the authors propose any explanation for this? Is there any supporting information in the literature.

Response 17: Thanks for the helpful comments. As we all know, fish deterioration will produce fishy taste, so, on the one hand, because the peptide pretreatment plays the role of anti-freeze protection, reducing the degree of deterioration of fish in the process of freeze-thaw cycle, thus maintaining the freshness and reducing the fishy taste, on the other hand, the peptides themselves have fresh flavor.

Point 18: (f) Section 3.3 – Is there any supporting information in the literature that can be cited to support the interpretation here?

Response 18: Thanks for the helpful comments. The references have been added in the manuscript.

Point 19: (g) Sections 3.4 and 3.5 – Are there any studies that can be cited to support the interpretation in the second half of these sections?

Response 19: Thanks for the helpful comments. The references have been added in the manuscript.

Point 20: (h) Section 3.8 – “Significantly, the disruption activated the oxidase, accelerated protein oxidation, and increased myofibril’s carbonyl group content.” – This part is unclear. What oxidase was the authors referring to? Also, would it be appropriate then to cite references to support the statement?

Response 20: Thanks for the helpful comments. This phenomenon can be attributed to the formation of ice crystals during freezing that causes cell rupture, leading to the release of proteases such as cathepsins, matrix metalloproteinase (MMP), and oxidases such as cytochrome oxidase and lipoxygenase and pro-oxidants including reactive oxygen speciesspeeding up the oxidation of fatand protein. This reference has been added at the end of this paragraph in the manuscript.

Point 21: Section 3.9 – “As depicted in Fig. 5(d), the dityrosine content of FF, Control, TH, CP and TH-CP was 1102.7, 1674, 1533.3, 1253.3, 1284 and 1406.7 AU, respectively.” – This part is unclear. Please recheck why there would be six values, not five?

Response 21: Thanks for the helpful comments. It has been checked and corrected in the manuscript.

Point 22: (j) Section 3.12 – “The contrast in both TP and CP was not statistically significant (P<0.05).” – If it is not statistically significant, it should be P > 0.05, instead. Please recheck/revise.

Response 22: Thanks for the helpful comments. It has been checked and corrected in the manuscript.

Point 23: CONCLUSIONS: Please consider whether it is appropriate/accurate to use the term “peptide” here, e.g., “The P. crocea peptide prepared by trypsin…” and “… the peptide effectively delayed the oxidation”. Since what the authors actually prepared were peptide mixtures or hydrolysates, calling the samples “peptide’ might mislead the readers into thinking pure/purified (single) peptides were used/prepared.

Response 23: Thanks for the helpful comments. It has been corrected as “trypsin hydrolysate of P. crocea protein” in the manuscript.

Reviewer 3 Report

The main question that is addressed by Xu et al. is the multiple mechanisms underlying the cryoprotective activity of P. crocea hydrolysates. The topic is considered original as it investigates significant issues and parallel mechanisms that are required to protect tissue from freeze and thaw damage. Specific improvements are required in the literature survey both in the introduction and discussion which are provided as extra comments. The conclusion needs to be revised as it shows redundancy with the abstract.

Other comments:

1-     The study should emphasize in the background what are the current limitations of other antifreeze peptides and the necessity to investigate new candidates

2-     How long is the half-life of the current peptides for protection compared to the conventional antifreeze agents?

3-     Several sections in the materials and methods could be merged together to avoid extra subsection creation.  The Bradford test details are not required. On the other hand sections 2.10.2, 2.10.3, and 2.10.4 need a brief description.

4-     There are several sentences in the methods that are written in an imperative form which is not appropriate for the article such as “…Weigh 2 g of the tested sample, and add 25 mL of pure water, homogenized for 1 min…”

5-     The discussion needs to be enriched by a strong literature survey to highlight the importance of the current findings. I propose to separate the discussion from the results for this purpose.

Author Response

Response to Reviewer 3 Comments

The main question that is addressed by Xu et al. is the multiple mechanisms underlying the cryoprotective activity of P. crocea hydrolysates. The topic is considered original as it investigates significant issues and parallel mechanisms that are required to protect tissue from freeze and thaw damage. Specific improvements are required in the literature survey both in the introduction and discussion which are provided as extra comments. The conclusion needs to be revised as it shows redundancy with the abstract.

The authors corrected the manuscript as suggested. Hence, the paper is ready for publication after minor revision, as follows:

Point 1: The study should emphasize in the background what are the current limitations of other antifreeze peptides and the necessity to investigate new candidates.

Response 1: Thanks for the helpful comments. the current limitations of other antifreeze peptides are that most of the antifreeze peptides on the market are of plant origin such as winter wheat antifreeze peptides, but they require high purity when applied to aquatic products, so there is a lack of research focusing on fish-derived antifreeze peptides applied to frozen storage of fish products. This point is expressed in the last sentence of the second paragraph in INTRODUCTION in the revised manuscript.

Point 2: How long is the half-life of the current peptides for protection compared to the conventional antifreeze agents?

Response 2: Thanks for the helpful comments. Because the three freeze-thaw cycles [1] are accelerated experiments, the half-life of traditional antifreeze could not be compared. In this study, compared with the addition of 4% sucrose [2], one of the traditional antifreeze, the trypsin hydrolysate showed good antifreeze protection activity during the whole process. Therefore, the half-life of trypsin hydrolysate is longer than that of commercial antifreeze sucrose.

[1] Du X., Chang P., Tian J., Effect of ice structuring protein on the quality, thermal stability and oxidation of mirror carp (Cyprinus carpio L.) induced by freeze-thaw cycles. Lwt, 2020, 124: 109140.

[2] Donald, G. A. M.; Lanier, T. C., Actomyosin Stabilization to Freeze-Thaw and Heat Denaturation by Lactate Salts. Journal of Food Science 1994, 59 (1), 101-105.

Point 3: Several sections in the materials and methods could be merged together to avoid extra subsection creation.  The Bradford test details are not required. On the other hand sections 2.10.2, 2.10.3, and 2.10.4 need a brief description.

Response 3: Thanks for the helpful comments. All have been modified accordingly in the revised manuscript.

Point 4: There are several sentences in the methods that are written in an imperative form which is not appropriate for the article such as “…Weigh 2 g of the tested sample, and add 25 mL of pure water, homogenized for 1 min…”.

Response 4: Thanks for the helpful comments. The relevant details have been corrected in the revised manuscript.

Point 5: The discussion needs to be enriched by a strong literature survey to highlight the importance of the current findings. I propose to separate the discussion from the results for this purpose.

Response 5: Thanks for your comments. The discussion part has been enriched in the revised manuscript. The format of the journal Foods is written by mixing results with discussions [1-3]. The references cited in the results and discussion part have provided the detailed supports about these findings. The combination of results and discussion may be more targeted to explain the importance of each finding.

[1] Sherif M., Eman M., Ralf G., at al. Comparative Untargeted Metabolic Profiling of Different Parts of Citrus sinensis Fruits via Liquid Chromatography–Mass Spectrometry Coupled with Multivariate Data Analyses to Unravel Authenticity, Foods 2023, 12(3), 579. https://doi.org/10.3390/foods12030579.

[2] Weifeng Gao, Ye Yuan, Zhi Huang, at al. Evaluation of the Feasibility of Harvest Optimisation of Soft-Shell Mud Crab (Scylla paramamosain) from the Perspective of Nutritional Values, Foods 2023, 12(3), 583. https://doi.org/10.3390/foods12030583.

[3] Daria Gmižić, Marija Pinterić, Maja Lazarus, Ivana Šola, High Growing Temperature Changes Nutritional Value of Broccoli (Brassica oleracea L. convar. botrytis (L.) Alef. var. cymosa Duch.) Seedlings, Foods 2023, 12(3), 582. https://doi.org/10.3390/foods12030582.

Round 2

Reviewer 1 Report

The questions were well answered.

Reviewer 3 Report

Notes to be considered for further manuscripts

It is expected that responses would be highlighted in color in the revised manuscript.

Similarly, the points that are added to the revised manuscript should be repeated in the response form.